

# Electrical characteristics of the extracellular fluid in the body segments of *Apis mellifera* bees

Juan Hernandez[1], Fredy Mesa[2], Anderson Dussan[3] and Andre Riveros[1]

[1] Faculty of Natural Sciences, Universidad del Rosario, Bogotá, Cundinamarca, Colombia
[2] Faculty of Engineering and Basic Sciences, Fundación Universitaria Los Libertadores, Bogotá, Cundinamarca, Colombia
[3] Physics Department, Universidad Nacional de Colombia, Bogotá, Cundinamarca, Colombia

## ABSTRACT

This study investigates the electrical properties of the extracellular fluid in honeybees (*Apis mellifera*) and its relationship with different body segments. By characterizing resistance, capacitance, and electrical impedance, aspects such as ionic composition, molecular polarization, and the differential response of live bees to electrical stimuli were evaluated. The results show that electrical characteristics vary significantly depending on the body segment, with the head exhibiting high resistance values and the abdomen displaying high capacitance, reflecting differences in molecular composition and functionality. Additionally, experiments with live bees demonstrated the feasibility of measuring electrical parameters non-invasively, opening new possibilities for monitoring the health of these pollinators under controlled conditions and in natural environments. This work lays the foundation for developing innovative tools in ecological monitoring, the assessment of environmental stressors, and the sustainable management of bee colonies.

## INTRODUCTION

The electrical properties of biological materials (*Heng et al., 2023*; *Kuang & Nelson, 1998*; *Leijsen et al., 2021*; *Miklavčič, Pavšelj & Hart, 2006*; *Sasaki et al., 2022*; *Smolyanskaya et al., 2018*) have been extensively studied due to their potential applications in biotechnology (*Atkinson et al., 2023*; *Bedi et al., 2022*; *Forro et al., 2021*), diagnostics (*Anushree et al., 2022*; *Russo et al., 2022*), and physiological monitoring. Biological tissues and fluids exhibit electrical behaviors (*Angenent et al., 2024*; *Jalilinejad et al., 2023*; *Joshi, Mishra & Narayan, 2021*), which can be explored to understand their composition, structure, and functionality. The characterization of these electrical properties provides insights into ion transport, membrane dynamics, and cellular interactions. For instance, dielectric spectroscopy has been applied in mammalian tissues to study hydration levels and cellular density, in plant tissues to assess water content and ion fluxes, and in microbial systems to monitor metabolic activity. In these systems, electrical resistance and capacitance measurements have revealed information about ion mobility, membrane integrity, and the behavior of polar molecules (*Gabriel, Gabriel & Corthout, 1996*; *Heileman, Daoud &*

Corresponding author
Juan Hernandez,
juanp.hernandez@urosario.edu.co

*Tabrizian, 2013*; *Yao et al., 2020*). Despite this, insect extracellular fluids remain understudied, especially in ecologically relevant species such as honeybees. Exploring their electrical properties may uncover useful biomarkers of physiological condition and environmental stress.

In the case of insects, particularly honeybees (*Apis mellifera*), the electrical characteristics of their biological fluids remain largely unknown. Their role as pollinators makes their health and physiological status topics of great interest. Recent studies have emphasized the importance of monitoring their physiological and behavioral responses to environmental stressors (*Hung & Yiin, 2023*; *Mayack et al., 2022*; *Woodard, 2017*), such as pesticide exposure (*Li et al., 2022*; *Lourenço et al., 2021*; *Shaher & Manjy, 2020*) and dietary changes (*Ardalani et al., 2021*; *Bryś, Skowronek & Strachecka, 2021*). However, the electrical characteristics of their extracellular fluids have received little attention, despite their potential to serve as indicators of ionic composition and metabolic activity.

The extracellular fluid of honeybees plays a fundamental role in maintaining homeostasis (*Bournonville et al., 2023*; *Gábor et al., 2017*; *Richardson et al., 2018*), transporting nutrients (*Kannan et al., 2024*; *Li et al., 2024*; *Tafi et al., 2024*), and supporting immune responses (*Gábor et al., 2017*; *Mallon, Brockmann & Schmid-Hempel, 2003*). The electrical properties of this fluid, such as resistance, capacitance, and impedance, may offer new avenues for studying the physiological state of bees under various conditions. Additionally, the ability to evaluate current-voltage relationships in the thorax of live bees could provide information about ion mobility and potential interactions with the surrounding tissues.

This study aims to investigate the electrical characteristics of the extracellular fluid of honeybees under laboratory conditions. Four experiments were conducted: three focusing on the extracellular fluid and one involving live bees. These experiments determined the conductivity, capacitance, and impedance of the extracellular fluid. Finally, current-voltage measurements allowed for the characterization of the extracellular fluid in the thorax of immobilized live bees. The experiments sought to establish a foundation for evaluating the electrical properties of honeybee fluids, offering potential applications in physiological monitoring and environmental assessments. This research provides a novel perspective on bee health and their electrical interactions.

## MATERIALS AND METHODS

### Experimental animals

The honeybees (*Apis mellifera*) used in this study were collected from the apiary at Universidad del Rosario (Bogotá, Colombia). The insects selected were foragers captured during their morning activity (~9 a.m). The bees were housed in disposable cups modified to allow ventilation and feeding. Immediately after collection, the insects were transported to the laboratory, where temperature, relative humidity, and light type were controlled ($T = 32 \pm 0.2\ °C$; $RH = 52 \pm 5\%$; $\lambda = 750\ nm$). Red light (750 nm) was used to minimize behavioral alterations during handling, as bees exhibit reduced visual sensitivity at this wavelength, allowing more consistent physiological measurements. The bees were kept under these conditions for 24 h, during which they were fed with a 1 M sucrose solution

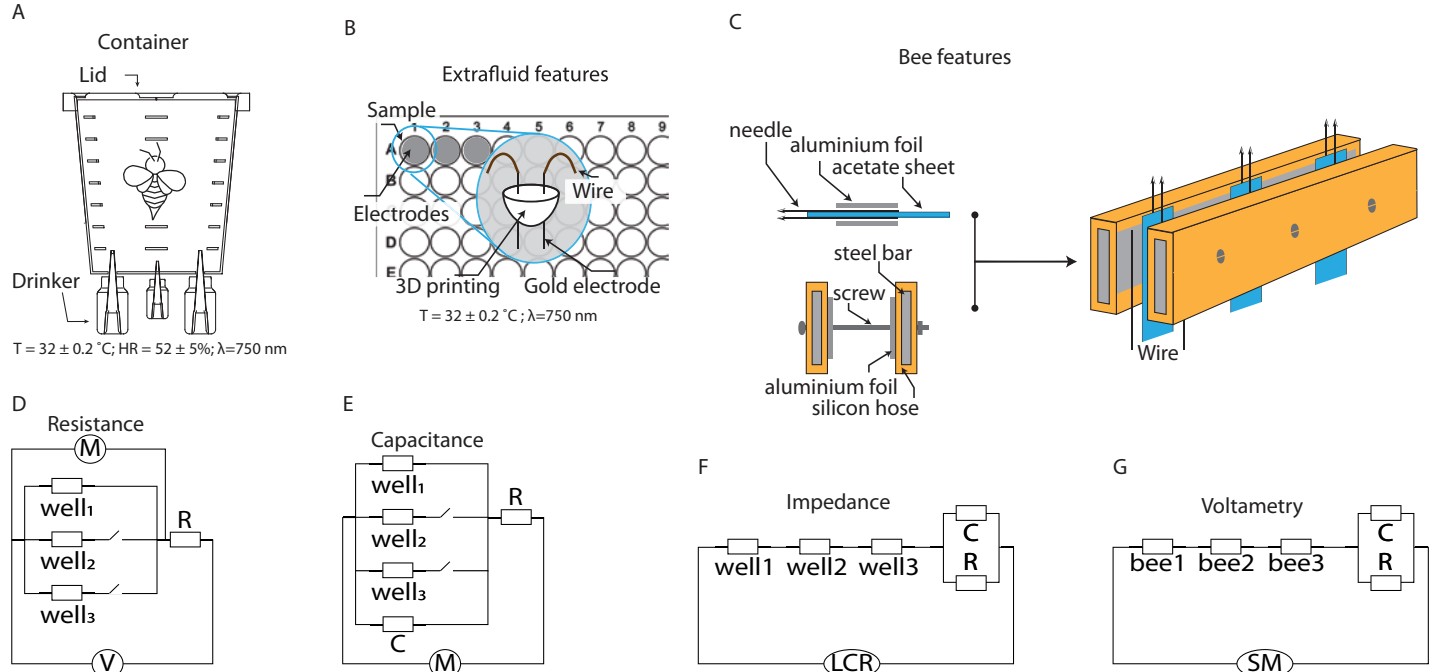

**Figure 1 Schematics of the setups for characterizing extracellular fluid and live bees.** (A) Container for maintaining bees under laboratory conditions. (B) System for characterizing the extracellular fluid of bees. (C) System for characterizing the extracellular fluid of live bees. (D–G) Equivalent circuits for modeling the electrical characteristics of the extracellular fluid.

(33.1% w/w) (Fig. 1A). Bee maintenance protocols followed the Organisation for Economic Co-operation and Development (OECD) guidelines for insecticide exposure (*OCDE 245, 2017*).

## Extracellular fluid extraction

The extraction of extracellular fluid was performed after euthanizing the bees using low temperatures (*Kovac et al., 2014*; *MacMillan & Sinclair, 2011*). For this, the insects were placed in a cooling chamber for 24 h. Subsequently, the body segments (head, thorax, and abdomen) were placed in filtered micropipette tips, and the fluid was extracted *via* centrifugation at 4 °C. Each extraction involved seven bees subjected to 2,000 relative centrifugal force (RCF) for 3 min. To preserve the extracellular fluid, 500 μL of distilled water was added. Finally, the samples were stored at −20 °C until analysis.

## Devices for electrical characterization

The first device used for electrical characterization was constructed using 96-well cell culture plates, cylindrical gold electrodes (1.0 mm diameter) mounted on a 3D-printed structure. The design ensured that the electrodes were positioned 3 mm apart, and their terminals allowed for secure attachment to the plate by pressure. Wires were soldered to the electrodes to facilitate manipulation on a breadboard, enabling series or parallel configurations depending on the desired characterization (Fig. 1B).
The second device used for characterization consisted of stainless-steel electrodes and acetate sheets. The electrodes were built using 32G hypodermic needles attached to each side of acetate sheets (2 cm × 4 cm × 0.02 cm). The acetate thickness ensured the separation distance between the two needles. Aluminum foil was added to each side of the acetate to increase the conductive area of the needles. This assembly was connected to steel rods covered with silicone tubing, onto which aluminum foil was attached to ensure electrical conduction. Mechanical stabilization was achieved using screws and nuts (Fig. 1C).

For live characterizations, plastic holders were constructed to immobilize bees, facilitating electrical measurements. The holders were made using 1,000 µL micropipette tips, which were modified with longitudinal grooves to allow the bees to slide in. The insect's abdomen remained inside the tube to prevent stings and facilitate handling. The bee's thorax was secured to the tube using a small piece of paraffin paper, immobilizing the bees for characterization.

## Experiment 1: electrical resistance of the extracellular fluid from bee body segments

This experiment was designed to address questions related to the resistive behavior of the extracellular fluid in honeybee body segments. Additionally, electrical resistance measurements aimed to explore the electrical properties of the samples from each body segment.

The experiment involved measuring the electrical resistance of extracellular fluid samples from each body segment of the bees. For this, 30 µL samples were placed in the wells of a cell culture plate. The experiment evaluated configurations of 1, 2, and 3 wells, which were connected in parallel. Gold electrodes were inserted into the wells, and their cables were connected in series with a 2,200 Ω electrical resistor. These configurations were connected to a digital multimeter (DMM7500 Keithley) to measure the resulting voltage drop across the samples. The instrument was set to acquire 200 data points per minute. To ensure measurement accuracy, five repetitions were performed.

## Electrical model used for resistance measurements

Electrical characterization was performed using a voltage divider circuit, a configuration that distributes the voltage from a power source among two or more resistive elements connected in series. This method is widely used in electronic applications to obtain a lower voltage from a higher voltage source. A voltage divider consists of a voltage source ($Vin$) that provides the total voltage, which is then distributed among the circuit components. The resistive materials are connected in series with the source, and the potential difference ($Vout$) is measured across the components (Fig. 1D). Specifically, in the proposed configuration, the electrical resistances of the extracellular fluid samples ($Rsample$) were connected in series with a known resistor ($R$), following the relationship:

$$R_{sample} = -\frac{V_{out}/V_{in}\,R}{(V_{out}/V_{in} - 1)}. \tag{1}$$

## Experiment 2: electrical capacitance of the extracellular fluid from bee body segments

This experiment was designed to address questions related to the capacitive behavior of the extracellular fluid in honeybee body segments. Additionally, the samples from each body segment were analyzed to determine their dielectric characteristics in terms of electrical capacitance.

The experiment involved measuring the electrical discharge of a capacitor connected in parallel with the extracellular fluid samples from different body segments. For this, 30 µL samples were placed in the wells of a cell culture plate. The experiment evaluated configurations of 1, 2, and 3 wells, which were connected in parallel. Gold electrodes were inserted into the wells, and their cables were connected in parallel with a 224 nF ceramic capacitor. This configuration was then connected in series with a 5 MΩ resistor. The capacitor was charged using a voltage source set to a constant value of 0.9 V. The discharge measurements were performed by recording the voltage drop using a digital multimeter (DMM7500 Keithley), which was configured to acquire 200 data points per minute. To ensure measurement accuracy, five repetitions were performed.

### Electrical model used for capacitance measurements

To analyze the electrical discharge of a capacitor, potential differences are applied to charge it, and then it is allowed to discharge through a resistor. This type of circuit is useful for studying how the energy stored in the capacitor dissipates through the resistor. In this study, the system components included the extracellular fluid samples ($C\ sample$), along with a capacitor ($C$) and a resistor ($R$) with known values. The resistor provided a discharge path, allowing the stored charge ($Q$) to leave the capacitor plates, resulting in a decrease in electric potential over time (Fig. 1E).

The voltage ($V$) across such a system should decrease exponentially with time ($t$), following the equation:

$$V = V_o e^{-t/\tau} \tag{2}$$

where $V_0$ is the initial voltage and $\tau$ is the time constant ($RC$). The constant $\tau$ determines the time required for the voltage to drop to approximately 36.8% of its initial value.

## Experiment 3: electrical impedance of the extracellular fluid from bee body segments

This experiment addressed questions related to the resistive and capacitive behavior of the extracellular fluid in honeybee body segments through impedance measurements. Additionally, electrical impedance measurements were incorporated to determine the resistance and reactance of extracellular fluid samples from each body segment.
The experiment involved measuring the reactance and electrical resistance of the extracellular fluid samples, which were connected to a capacitor and a resistor. For this, 30 µL samples were deposited in the wells of the culture plate, with only three wells evaluated, which were connected in series. Gold electrodes were inserted into the wells, and their cables were connected in series with a 224 nF ceramic capacitor and a 2,200 Ω resistor. Measurements were performed using an Inductance-Capacitance-Resistance (LCR) meter (E4980AL; Keysight, Santa Rosa, California, USA), configured to measure resistance and reactance five times for each frequency. The electrical excitation of the samples was selected across four frequency ranges to obtain impedance at both low and high frequencies: I1 = 20–120 Hz; I2 = 120–1,020 Hz; I3 = 1,020–10,020 Hz; I4 = 10,020–100,020 Hz. The LCR meter was programmed using the Command Expert software from Excel, and to ensure measurement accuracy, each measurement was repeated five times.

## Electrical model used for impedance measurements

Electrical impedance ($Z$) is a generalized measure of the opposition that a circuit or sample presents to the flow of current when an alternating potential difference is applied. This measure of electrical resistance for circuits with voltages varying at different frequencies ($\omega = 2\pi f$) includes both resistance ($R$) and reactance ($X$). Impedance is a complex quantity ($j = \sqrt{-1}$), indicating that it consists of a real and imaginary component ($Z = R + Xj$). The real component represents the opposition to current flow due to energy dissipation, while the imaginary component represents the opposition to current flow due to reactive elements such as capacitors (Fig. 1F).

In systems exhibiting both capacitive and resistive behavior, as observed in the extracellular fluid samples, impedance follows the form:

$$Z = \frac{R}{1 + R^2\omega^2 C^2} + \frac{R^2\omega C}{1 + R^2\omega^2 C^2}j. \tag{3}$$

The limit of $Z$ as the frequency approaches zero represents the resistive characteristic of the system; however, as the frequency approaches infinity, the impedance value becomes zero.

## Experiment 4: electrical characteristics of live bees

The questions addressed in this section focused on the electrical response observed in the thorax due to electrical stimuli. The electrical responses of the bee thorax demonstrated the resistive and capacitive behavior of living tissue.

The experiment involved obtaining current *vs.* voltage (I–V) curves from live bees, which were immobilized and connected in parallel to a resistor-capacitor (RC) electrical circuit using hypodermic needles. Measurements were performed using a source meter (Keithley 2450), configured to produce electrical excitation and record the system's electrical current over three cycles. The excitation potential difference range was set from −900 to 900 mV, with a sweep rate of 18 mV/s. The capacitance exhibited by the thorax was calculated from the data obtained from the I–V curves.

## Electrical model used for impedance measurements

Voltammetry is a set of electrical techniques used to study the characteristics of a system by measuring the electrical current generated when a potential is applied to a solution. The potential (voltage) applied to the electrodes serves to excite the mobile components of the samples, with the generated currents being proportional to the electroactive species in the solution. In linear sweep voltammetry, a potential that varies linearly with time is applied while the current is measured. The applied potential can be specified within intervals to perform forward and reverse sweeps, providing information on reversible and non-reversible processes. The relationship between current and potential is represented in a voltammogram, where the horizontal axis displays the applied potential, and the vertical axis shows the recorded current. When the samples exhibit both capacitive and resistive characteristics, the current ($I$) in the system will increase or decrease depending on the applied potential (Fig. 1G). The system's variations will be proportional to the applied voltage, a behavior described by a composite equation for each of the components:

$$I = C\frac{dV}{dt} + \frac{V}{R}. \tag{4}$$

The first component of Eq. (4) corresponds to the capacitive current, while the second is related to the electrical resistance of the system.

## RESULTS

### Experiment 1: electrical resistance of extracellular fluid by body segment

In the electrical resistance experiment, 105 bees were sacrificed to obtain extracellular fluid from each of their body segments. The highest potential differences were observed when using a single well, with the distribution range remaining constant over the evaluation period [median; quartile 1; quartile 3]: [740 mV; 736 mV; 743 mV] head; [698 mV; 694 mV; 701 mV] thorax; [678 mV; 674 mV; 682 mV] abdomen. The samples exhibited intermediate potential values when two wells were used; however, the dispersion of values increased compared to measurements with a single well [median; quartile 1; quartile 3]: [635 mV; 628 mV; 641 mV] Head; [579 mV; 571 mV; 586 mV] thorax; [554 mV; 546 mV; 561 mV] abdomen. In contrast, the values obtained with three wells were the lowest, and a growing dependence over time was also observed [median; quartile 1; quartile 3]: [564 mV; 554 mV; 573 mV] head; [503 mV; 491 mV; 511 mV] thorax; [477 mV; 465 mV; 485 mV] abdomen (Fig. 2A).

Additionally, resistance estimations using Eq. (1) indicated that the electrical characteristics changed depending on the number of wells and the body segment. In all cases, the samples from the head segment exhibited the highest electrical resistance values, in contrast to the fluid obtained from the abdomen. The observed resistances in the different segments decreased as the number of wells increased. The percentage differences observed with two and three wells, relative to the first well, remained consistent (mean ± SD): $47.77 \pm 0.21\%$ (well1–well2); $63.33 \pm 0.23\%$ (well1–well3). These results indicated that as the number of wells increased, the system behaved similarly to a parallel circuit
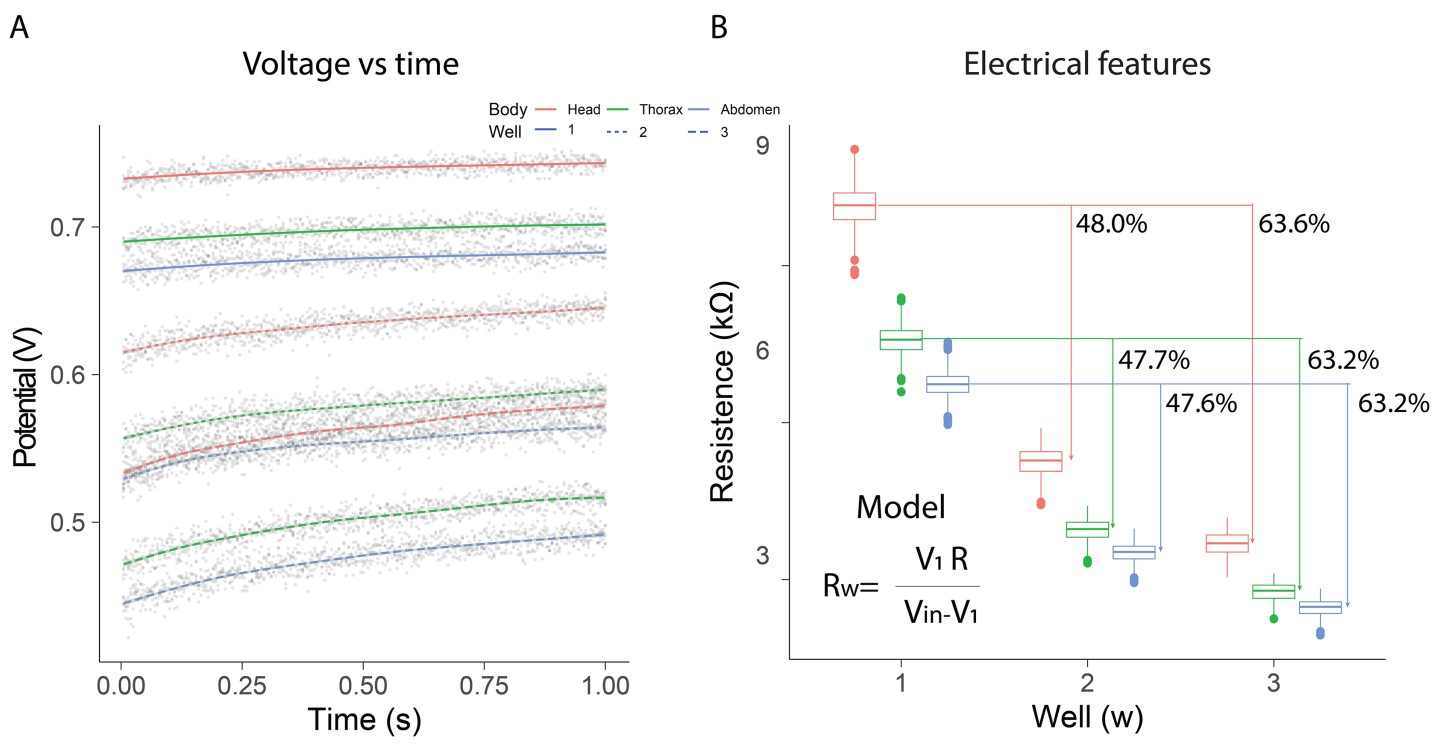

**Figure 2 Electrical resistance of the extracellular fluid.** (A) Potential difference observed over one minute for the fluid obtained from bee body segments. (B) Estimation of electrical resistance for each segment based on the number of wells.

**Table 1 Electrical resistance (kΩ) of extracellular fluid by body segment and well configuration.** Values are shown as median [Q1–Q3].

| Body segment | Resistance 1 well | Resistance 2 wells | Resistance 3 wells |
|---|---|---|---|
| Head | 10.2 [9.9–10.4] | 5.3 [5.1–5.4] | 3.7 [3.5–3.9] |
| Thorax | 7.6 [7.4–7.8] | 4.0 [3.8–4.1] | 2.8 [2.6–2.9] |
| Abdomen | 6.7 [6.6–6.9] | 3.5 [3.4–3.6] | 2.5 [2.4–2.6] |

(Fig. 2B). Consequently, the extracellular fluid samples exhibited resistive behavior to the flow of electrical current. The resistance values measured for each condition are summarized in Table 1. The median and interquartile ranges show a consistent pattern across body segments and well configurations, reinforcing the segmental differences in electrical resistance observed in this study.

## Experiment 2: electrical capacitance of extracellular fluid by body segment

In the electrical capacitance experiment, 105 bees were sacrificed to obtain extracellular fluid from each of their body segments. The potential of the samples connected to the capacitor varied similarly to a discharge circuit, with an initial potential of 0.86 ± 0.02 V. The 50% discharge (0.43 ± 0.01 V) was achieved within 0.18 s, while the 36.6% discharge (0.31 ± 0.01 V) was reached in 0.27 s. In the 36.6% discharge interval, significant

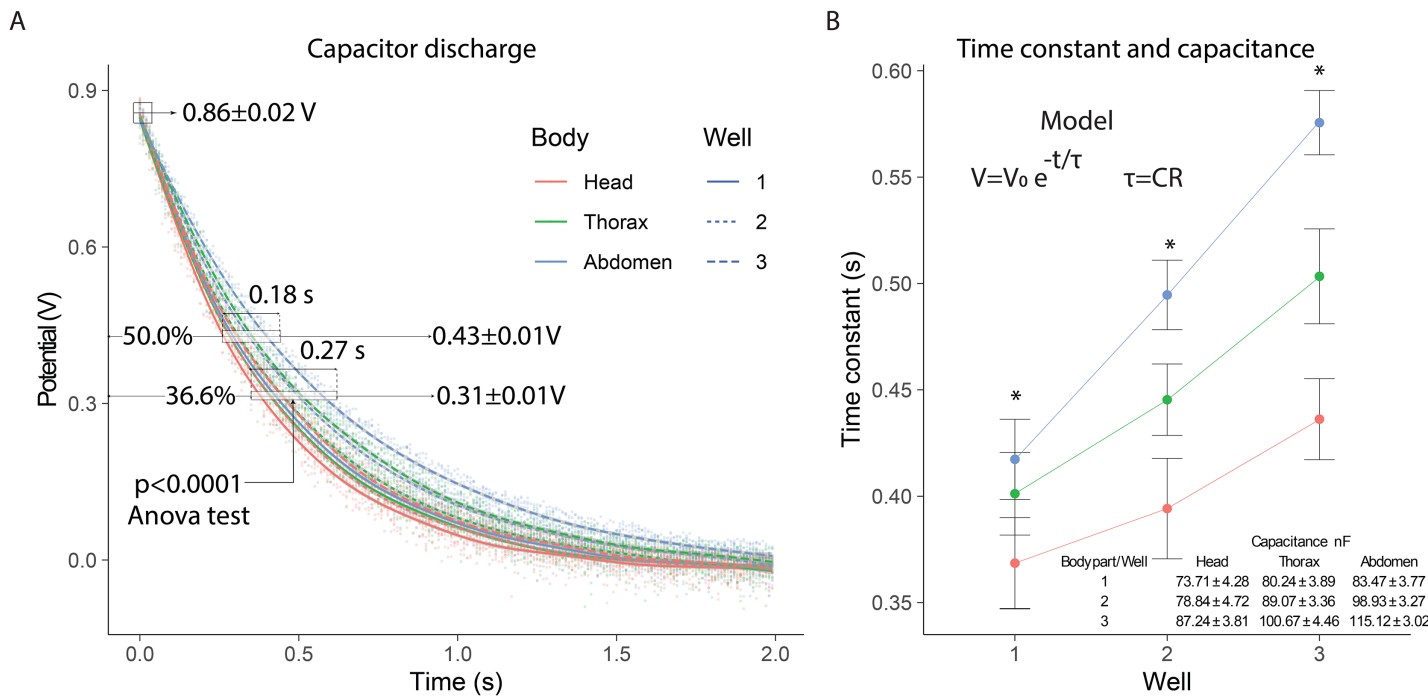

**Figure 3 Electrical capacitance of the extracellular fluid.** (A) Observed discharge potential difference of extracellular fluid samples from bees. (B) Estimation of the time constant and capacitance based on the number of wells. Asterisks indicate statistically significant differences between well configurations (1, 2, and 3 wells) within each body segment (one-way ANOVA, $p < 0.0001$; Tukey's test).

differences ($p < 0.0001$) were observed in the discharge potentials for each body segment depending on the number of wells used: ($F_{2,357} = 28$) head; ($F_{2,357} = 53$) thorax; ($F_{2,357} = 108$) abdomen (Fig. 3A).

In the 0.35 to 0.62 s interval, the 36.6% capacitor discharge was achieved, showing variations in the time constant for each body segment. The time constants exhibited significant differences ($p < 0.0001$) depending on the number of wells evaluated: ($F_{2,357} = 16.5$) well 1; ($F_{2,357} = 47.0$) well 2; ($F_{2,357} = 72.1$) well 3. The time constants were highest for the abdomen, intermediate for the thorax, and lowest for the head, with greater distinction as the number of wells increased. This consecutive increase in the time constant indicated that the system's capacitance also increased with the number of wells, with values in the nanofarad range (Fig. 3B).

The capacitance measurements showed consistent differences between body segments. The abdominal segment presented the highest capacitance values, followed by the thorax and the head. This trend was maintained across all well configurations. The values obtained indicate that the extracellular fluid can store electrical charge, and its capacitance is affected by anatomical and physiological differences between segments. The capacitance values recorded in the abdominal segment were consistently higher than those from the thorax and head. This may be due to the higher content of polar molecules and metabolites in the abdomen, increasing its ability to store charge. These values are comparable, in order of magnitude, to those observed in tissues with high dielectric properties such as skeletal

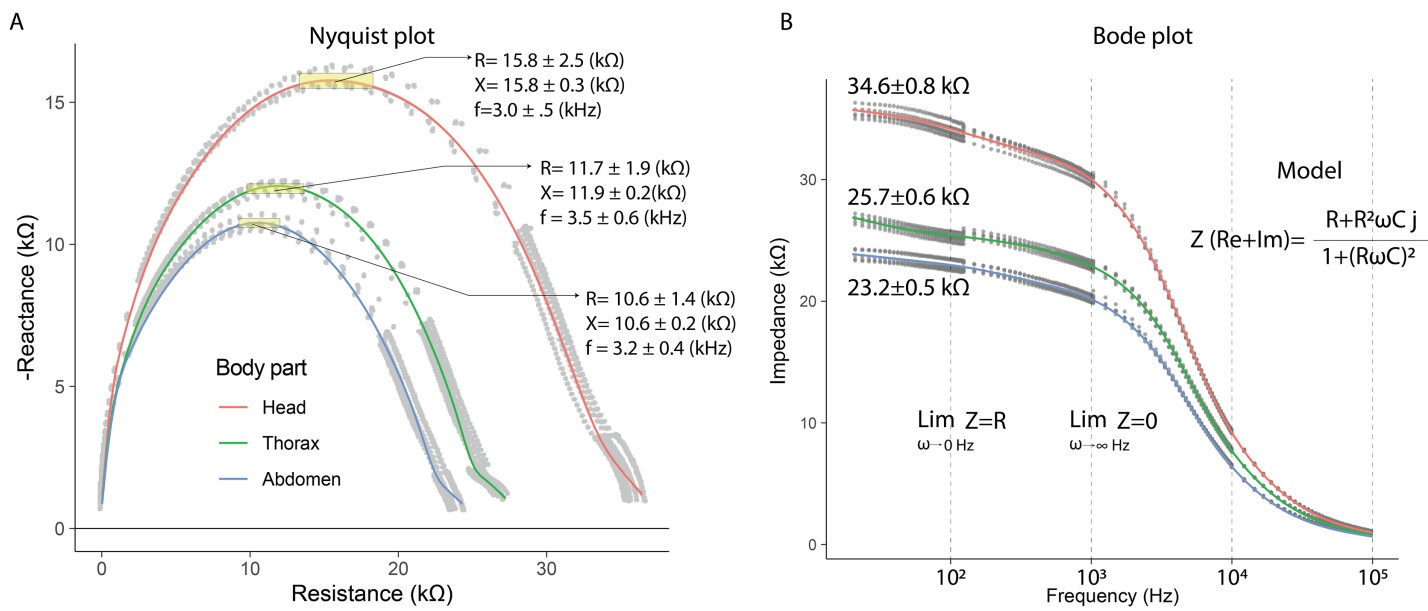

**Figure 4 Capacitive and resistive behavior of extracellular fluid samples.** (A) Nyquist plot showing the relationship between reactance and resistance in the electrical system. (B) Bode plot showing the relationship between electrical impedance and the excitation frequency of the sample.

muscle or lipid-rich membranes (*Zimmermann & Van Rienen, 2021*; *Gabriel, Gabriel & Corthout, 1996*).

## Experiment 3: electrical impedance of extracellular fluid by body segment

In the electrical capacitance experiment, 35 bees were sacrificed to obtain extracellular fluid from each of their body segments. The electrical impedance measurements of the extracellular fluid exhibited frequency-dependent electrical characteristics. This was evidenced in the Nyquist plots, where the relationship between reactance (imaginary component) and electrical resistance (real component) displayed curved segments, indicating both capacitive and resistive behavior. The Nyquist plot showed that the curves corresponding to each body segment had a circular geometry, with the largest radius observed for the head, followed by the thorax, and lastly, the abdomen.

The Nyquist plot also revealed that reactance and resistance values varied depending on the body segment, with the highest values found in the head, followed by the thorax, and the lowest in the abdomen. The head segment was characterized by the highest reactance and resistance values at frequencies of $3.0 \pm 0.1$ kHz (*mean $\pm$ SD*): $15.8 \pm 0.3$ k$\Omega$ (X); $15.8 \pm 2.5$ k$\Omega$ (R). In contrast, at $3.5 \pm 0.6$ kHz, thorax samples exhibited lower reactance and resistance values compared to the head (*mean $\pm$ SD*): $11.9 \pm 0.2$ k$\Omega$ (X); $11.7 \pm 1.9$ k$\Omega$ (R). Finally, the electrical characteristics of the abdomen samples showed further reductions at $3.2 \pm 0.4$ kHz (*mean $\pm$ SD*): $10.6 \pm 0.2$ k$\Omega$ (X); $10.6 \pm 1.4$ k$\Omega$ (R) (Fig. 4A). These results indicate that the components of the extracellular fluid are susceptible to interactions with electric fields.

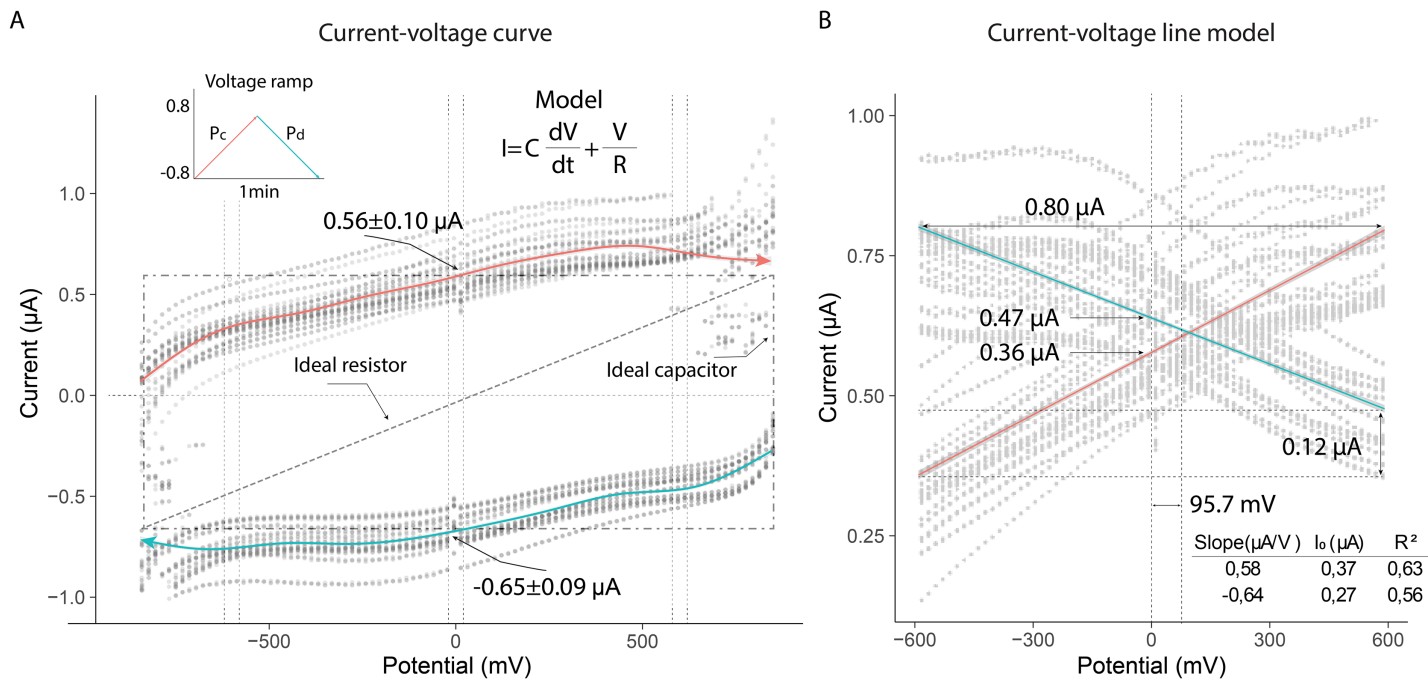

**Figure 5 Current-voltage curves of the thorax of live bees connected to a resistor and capacitor under uniform potential differences.** (A) Variation of current due to linear changes in electrical potential. (B) Linear model of the absolute values of electrical currents.

The Bode plot demonstrated that impedance decreased with increasing frequency, a characteristic behavior of systems that combine resistive and capacitive elements. In all cases, the electrical impedance of the extracellular fluid samples exhibited an inverse relationship with frequency (20 to 100,020 Hz), with this trend becoming more pronounced at higher frequencies. Impedance values were distinct for each body segment, with the highest values observed in the head, followed by the thorax, and the lowest in the abdomen. Specifically, at low frequencies (20 to 120 Hz), the resistive characteristics of the samples were predominant (*mean ± SD*): 34.6 ± 0.8 kΩ (Head); 25.7 ± 0.6 kΩ (Thorax); 23.2 ± 0.5 kΩ (Abdomen).

### Experiment 4: voltammetry of live bees

The current-voltage (*I–V*) curves showed that the thorax of the bees exhibited electrical characteristics under the experimental conditions. The linear changes in potential difference produced variable currents, which followed opposite trends depending on whether the potential was increasing (*Pa*) or decreasing (*Pb*). The behavior of both current curves reflected a dependence typical of systems that combine resistive and capacitive properties. These characteristics were evident when the electrical potential approached zero, generating nonzero currents in the system (*mean ± SD*): 0.56 ± 0.10 μA (*Pa*); −0.65 ± 0.09 μA (*Pb*) (Fig. 5A).

The electrical current in the thorax exhibited nearly linear behavior in the potential range from −600 mV to 600 mV (slope (μA/V); initial current (μA)): 0.56; 0.37 (*Pa*);

−0.64; 0.27 (*Pb*). The current across the thorax reached values close to 0.80 μA at the extreme potential values (−600 V; 600 V). Additionally, the currents generated at zero volts differed, indicating that the extracellular fluid in the thorax responded differently to the applied potentials: 0.47 μA (*Pb*); 0.36 μA (*Pa*). This result demonstrated that the composition of the fluid in the thorax induced a higher electrical resistance under decreasing potential conditions, leading to a lower current intensity (0.12 μA) (Fig. 5B).

## DISCUSSION

The results obtained in this study provide new insights into the electrical characteristics of extracellular fluids in honeybees (*Apis mellifera*) and their associated physiological implications. This approach allowed for the exploration of previously unstudied aspects, contributing to the development of precise tools for assessing pollinator health under different environmental conditions.

The experiments demonstrated that the electrical characteristics of extracellular fluid vary significantly depending on the body segment, as observed in other types of studies (*Beyenbach, 2016*; *Crailsheim, 1985*; *Strachecka et al., 2022*). In terms of electrical resistance, the samples from the head segment exhibited the highest values, while those from the abdomen showed the lowest. The differential resistance values of each segment may be due to the volume of fluid they contain and the electrical properties of their components. First, the amount of hemolymph in the head is lower than in the other two segments, as shown in previous studies. Additionally, the resistance gradient found suggests differences in ionic composition (sodium, chloride, and potassium) and in the dynamics of the tissues surrounding the extracellular fluid.

Regarding capacitance, the samples exhibited behavior consistent with capacitive systems during electrical discharge phases. This suggests that extracellular fluid samples exhibited electrical polarization effects when exposed to potential differences, indicating the presence of dielectric material within them (*Di Meo et al., 2022*; *Gabriel, Gabriel & Corthout, 1996*; *Zimmermann & Van Rienen, 2021*). This polarization-susceptible material likely corresponds to molecules with dipolar properties, which align with the field direction in the presence of an electric field. Notably, the fluid extracted from the abdomen exhibited the highest electrical capacitance, suggesting a higher presence of molecules that underwent polarization. The increased capacitance in the abdomen is consistent with the diverse elements present in this segment, such as water, pollen, sugars, and fats.

Compared to previously reported values in other biological fluids, the electrical resistance and capacitance measured in honeybee extracellular fluid fall within the expected range. For instance, hemolymph in lepidopteran insects such as *Spodoptera littoralis* and *Lymantria dispar* has demonstrated osmolarities and ionic concentrations that correspond to electrical conductivities between 4 and 6 mS/cm (*Pannabecker, Andrews & Beyenbach, 1992*; *Smagghe & Van Leeuwen, 2004*). These values are compatible with resistances in the 5–12 kΩ range, depending on developmental stage and solute content, and are consistent with the segmental values observed here for *Apis mellifera*. Likewise, capacitance values in the nanofarad range have been reported in tissues rich in

water and dipolar molecules, such as muscle fibers and hemolymph, due to their dielectric behavior (*Gabriel, Gabriel & Corthout, 1996*; *Zimmermann & Van Rienen, 2021*). The segmental pattern observed in honeybees—with higher resistance in the head and higher capacitance in the abdomen—reflects known physiological differences in fluid volume, metabolite composition, and compartmentalized function across body regions. These results support the idea that electrical measurements can serve as reliable indicators of physiological specialization in insect compartments.

An interesting observation in both resistance and capacitance measurements was the signal amplification with the addition of wells for measurements, behaving as passive electrical elements. In the case of electrical resistance, the addition of wells increased electrical effects, resembling a parallel resistor connection, where the total resistance is always lower than its components. For electrical discharge connections, the parallel configuration behaved as the sum of capacitances, exhibiting a greater ability to store electrical charge, similar to that observed in other tissues (*Schwan & Kay, n.d.*; *Sohn et al., 2000*; *Tsai et al., 2020*; *Yao et al., 2020*). These findings indicate that the evaluated biological samples can be accurately characterized using electrical resistance and capacitance models.

Electrical impedance also exhibited frequency-dependent behavior, with resistive characteristics predominating at low frequencies and capacitive behavior emerging at higher frequencies, similar to what has been observed in other biological systems (*Abasi et al., 2022*; *Grossi & Riccò, 2017*; *Leitzke & Zangl, 2020*; *Mesa et al., 2021*; *Wu et al., 2021*; *Yao et al., 2020*). This phenomenon aligns with systems containing mobile components (ions) and polarizable molecules (dipoles) (*Gabriel, Peyman & Grant, 2009*; *Gun, Ning & Liang, 2017*; *Heileman, Daoud & Tabrizian, 2013*; *Jaffrin & Morel, 2008*), as shown in the resistance and capacitance experiments. A novel finding from the impedance measurements was the dependence of reactance and resistance on excitation frequencies, indicating that frequencies between 100 and 10,000 Hz can effectively characterize extracellular fluid. Additionally, electrical impedance measurements allowed for the simultaneous characterization of both mobile and dielectric components of the extracellular fluid samples, similar to other biological systems (*Bedard et al., 2022*; *Jönsson et al., 2022*; *Veil et al., 2023*). Such measurements are highly relevant for assessing bee health, as they clearly differentiate each body segment. This approach could be particularly valuable for studies on particle segregation or retention due to xenobiotic exposure.

Finally, the experiments with live bees demonstrated that the thorax exhibited a differential electrical response when applying increasing and decreasing potentials. This result highlights the sensitivity of the thoracic extracellular fluid, potentially linked to the high metabolic and muscular activity in this region. The clear electrical current signal obtained from live bees is one of the strongest points of this discussion due to its potential real-world applications. As demonstrated, the current-voltage methodology required only a small number of bees to obtain data, and the bees were released after measurements. This suggests that electrical measurements could be standardized to characterize bees, enabling the collection of information to identify anomalies in their bodies. This could serve as a

Electrical characteristics of extracellular fluid in honey bee (3 wells)

| | Resistance kΩ | Capacitance nF | Impedance kΩ |
|---|---|---|---|
| Head | 3.7±0.2 | 87.2±3.8 | 34.6±0.8 |
| Thorax | 2.8±0.2 | 100.6±4.6 | 25.7±0.6 |
| Abdomen | 2.5±0.2 | 115.1±3.0 | 23.2±0.5 |

Electrical measurements could serve to identify physiological anomalies

**Figure 6 Electrical characteristics.** Segment-specific electrical characteristics of extracellular fluid in honeybees. The head shows high resistance and impedance but low capacitance, while the abdomen shows the opposite pattern. These patterns reflect physiological specialization and may serve as bio-markers of systemic health.

valuable tool for early diagnostics, helping prevent significant bee population losses due to a lack of health status information.

Although research on the electrical characteristics of biological fluids is well documented in other organisms, this study stands out for its focus on honeybees. The results confirm previously observed patterns in biological systems, such as the dependence of capacitance and impedance on fluid density and ionic composition. The application of these methodologies to insects is novel and opens a path for non-invasive physiological monitoring in bees.

The results of our measurements have significant implications both in biological and technological contexts. The electrical differentiation among body segments may be linked to specific functions, such as nutrient transport, immune activity, or metabolism. Additionally, the development of customized experimental devices facilitates precise electrical characterization, demonstrating their adaptability to complex and small biological systems. From an applied perspective, the electrical properties of extracellular fluid could be used to assess the impact of environmental stressors, such as pesticides or dietary changes, on bee health (Fig. 6). This has considerable potential for improving ecological monitoring practices and sustainable colony management.

One limitation of this study is the reliance on measurements under controlled conditions, which may limit direct applicability in natural environments. Additionally, the relatively small sample size may leave some aspects unexplored. Future research on electrical characteristics should examine variations in extracellular fluids due to different factors, including diet, environmental conditions, and pathologies. It would also be valuable to develop more non-invasive techniques to measure these properties in live bees, enabling more frequent monitoring with minimal impact on individuals. Thus, this study represents a significant step forward in understanding the electrical characteristics of biological fluids in honeybees and lays the foundation for future applications in physiological and environmental monitoring.

## CONCLUSIONS

This study not only expands the understanding of the electrical characteristics of biological fluids in honeybees (*Apis mellifera*) but also establishes a robust experimental framework for future research in physiological and environmental monitoring. The characterization of resistance, capacitance, and impedance in different body segments provides a tool to understand the biological responses of these essential pollinators to environmental challenges such as pesticide exposure and dietary changes.

Additionally, the integration of non-invasive techniques and customized devices highlights the potential of these methodologies for field applications, promoting a sustainable approach to colony management and preservation. This work represents a crucial step toward implementing strategies based on electrical parameters to ensure the health and survival of bees, which are fundamental pillars of ecosystems and global food security.

### Funding

This work was funded by the Universidad del Rosario and Fundación Universitaria Los Libertadores. The Universidad Nacional de Colombia supported this work *via* project QUIPU 202010042199, and Min-Ciencias through Conv. 937. The funders had no role in study design, data collection and analysis, decision to publish, or preparation of the manuscript.

### Grant Disclosures

The following grant information was disclosed by the authors:
Universidad del Rosario and Fundación Universitaria Los Libertadores.
The Universidad Nacional de Colombia: QUIPU 202010042199.
Min-Ciencias through Conv. 937.

### Competing Interests

The authors declare that they have no competing interests.

### Author Contributions

- Juan Hernandez conceived and designed the experiments, performed the experiments, analyzed the data, prepared figures and/or tables, authored or reviewed drafts of the article, and approved the final draft.
- Fredy Mesa conceived and designed the experiments, performed the experiments, analyzed the data, prepared figures and/or tables, authored or reviewed drafts of the article, and approved the final draft.
- Anderson Dussan performed the experiments, prepared figures and/or tables, authored or reviewed drafts of the article, and approved the final draft.

- Andre Riveros conceived and designed the experiments, analyzed the data, prepared figures and/or tables, authored or reviewed drafts of the article, and approved the final draft.

## Data Availability

The raw measurements are available in the Supplemental File.

## Supplemental Information

Supplemental information for this article can be found online at http://dx.doi.org/10.7717/peerj.19691#supplemental-information.

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
