# Peer review of "Electrical characteristics of the extracellular fluid in the body segments of Apis mellifera bees"

_PeerJ, doi:10.7717/peerj.19691_

## Round 0.1 · original submission · Minor Revisions

Thank you for your submission to PeerJ.

Please change your manuscript as the comments from the reviewers.

Reviewer 1 ·

Basic reporting

I'm not a specialist in the electrical properties of biological fluids as I only have experience dealing with electrical properties of solids. Consequently, my primary recommendation for this paper ties to that lack of knowledge on my part.

This ties to the very first paragraph..

"The electrical properties of biological materials (Heng et al., 2023; Kuang & Nelson, 1998; Leijsen et al., 2021; Miklav et al., n.d.; Sasaki et al., 2022; Smolyanskaya et al., 2018) have been extensively studied due to their potential applications in biotechnology (Atkinson et al., 2023; Bedi et al., 2022; Forro et al., 2021), diagnostics (Anushree et al., 2022; Russo et al., 2022), and physiological monitoring. Biological tissues and fluids exhibit electrical behaviors (Angenent et al., 2024; Jalilinejad et al., 2023; Joshi et al., 2021), which can be explored to understand their composition, structure, and functionality. The characterization of these electrical properties provides insights into ion transport, membrane dynamics, and cellular interactions."

please expand on what insights were gained, in what biological systems? This paragraph cites research but is very vague as to what the research says. This ties with the discussion..

Where the results gained from the paper are not tied to the research in general. Is the electrical resistance of the bee fluids high or low relative to the resistance of other biological fluids. Likewise with the capacitance.

Likewise " The experiments demonstrated that the electrical characteristics of extracellular fluid vary significantly depending on the body segment, as observed in other types of studies (Beyenbach, 2016; Crailsheim, 1985; Strachecka et al., 2022)." Do other electrical properties in other fluids vary by this much, or is this amount of variation unusual relative to other biological fluids? Some data needs to be presented and discussed from the cited papers to put the results found in bees in context. As written the paper presents the resistance and capacitance but I don't have any perspective on whether this is 'high', 'low', 'more or less variable' or what relative to other measurements in the literature. A figure or two would go a long way in presenting this..

Experimental design

Good

Validity of the findings

Good

Additional comments

Please put in context of greater literature.

Reviewer 2 ·

Basic reporting

Your manuscript is well written and easy to follow and understand.

Experimental design

Your description of the experimental design is also well written. I have some recommendations:
1. Please, explain why to use red light in the laboratory (line 77)
2. Please, use parentheses in Equation 1 to separate division from multiplication. In its present form is confuse

Validity of the findings

Your results are very interesting. I have some comments:
1. Please, report the median value of the voltage, not only the 1st and 3rd quartile.
2. Why you do not report the values calculated for the resistance? I consider much better to see the written value than the value in a graph

Additional comments

Your experimental protocol and results are very interesting. I feel is mising a discussion about the difference in electrical properties observed among the bee body parts. It is interesting to observe that there is a pattern from head to abdomen: from highest to lowest values or vice-versa. What could mean that tendency in biological terms? I consider you must comment more about that.

---

## Round 0.2 · Minor Revisions

Thanks for submitting your work to PeerJ.

I want to see your new version addressing the reviewers' comments.

Reviewer 1 ·

Basic reporting

Good

Experimental design

Good

Validity of the findings

Good

Additional comments

I have only one minor comment. Please, on Figure 6, include means +/- Standard deviations for the quantities discussed.

I'm happy with the manuscript, no need to see an additional draft.

Reviewer 2 ·

Basic reporting

The manuscript is ok. Good writing and easy to read.

Experimental design

The new version included the information lacking in the previous version. I consider the description of the experimental design is Ok.

Validity of the findings

The reported finding are very interesting. It deserves to be published in PeerJ.

Additional comments

I recommend the publication of the present manuscript. They need to correct two minor mistakes:
in line 204 correct j = (-1)^1/2
in line 402 and all the manuscript, scientific names in italic style.

---

## Round 0.3 · accepted · Accept

Congratulations!

Thank you for your submission to PeerJ.

Reviewer 1 ·

Basic reporting

I was happy with the previous manuscript with only very small changes - the authors completed the changes. Publish as is.

Experimental design

Publish as is.

Validity of the findings

Publish as is.

Additional comments

Publish as is.

Reviewer 2 ·

Basic reporting

No comment

Experimental design

No comment

Validity of the findings

No comment